

# An integrated modeling, verification, and code generation for uncrewed aerial systems: less cost and more efficiency

Jianyu Zhang[1], Long Zhang[2], Yixuan Wu[3], Linru Ma[2] and Feng Yang[2]

[1] School of Automation Engineering, University of Electronic Science and Technology of China, Chengdu, China
[2] National Key Laboratory of Science and Technology on Information System Security, AMS, Beijing, China
[3] School of Electronics and Information, Northwestern Polytechnical University, Xi'an, China

## ABSTRACT

Uncrewed Aerial Systems (UASs) are widely implemented in safety-critical fields such as industrial production, military operations, and disaster relief. Due to the diversity and complexity of implementation scenarios, UASs have become increasingly intricate. The challenge of designing and implementing highly reliable UASs while effectively controlling development costs and improving efficiency has been a pressing issue faced by academia and industry. To address this challenge, this article aims to examine an integrated method for modeling, verification, and code generation for UASs. This article begins to utilize Architecture Analysis and Design Language (AADL) to model UASs, proposing generic UAS models. Then, formal specifications describe a system's safety properties and functions based on these models. Finally, this article introduces a method to generate flight controller codes for UASs based on the verified models. Experiments demonstrate its effectiveness in pinpointing potential vulnerabilities in UASs during the early design phase and generating viable flight controller codes from the verified models. The proposed approach can also improve the efficiency of designing and verifying high-reliability UASs.

## INTRODUCTION

Uncrewed Aerial Systems (UASs) have become indispensable tools in many industries due to their enhanced processing capabilities, adaptability, and extended operational durations (*Gupta, Ghonge & Jawandhiya, 2013*; *Tan et al., 2020*). As systems adapt to different settings to become increasingly diverse, the complexity of UASs grows sharply (*Balestrieri et al., 2021*). Consequently, the safety and reliability of these systems turn out to be critical concerns.

UASs may encounter several security issues during operations, including hacking and component failures (*Mohsan et al., 2023*). Several security-enhanced technologies have been researched by both academia and industry to cope with these issues. This article categorizes these technologies into design-time security assurance and runtime security

Corresponding author
Long Zhang,
zhanglong10@nudt.edu.cn

assurance groups based on application stages. Design-time security assurance technologies are primarily applied during the design phase of UASs and include in-the-loop simulation testings (*Dai et al., 2021*), Failure Modes and Effects Analysis (FMEA) (*Huang et al., 2020*; *Shafiee et al., 2021*), and formal methods (*Clarke & Wing, 1996*). These technologies aim to identify and eliminate potential safety risks. System designs meeting safety requirements are also ensured. On the other hand, runtime security assurance technologies are mainly implemented during the operation of UASs and include real-time monitoring (*Witayangkurn et al., 2012*), autonomous decision-making and control (*Veres et al., 2011*), and Runtime Assurance (RTA) systems (*Schierman et al., 2020*; *Lee et al., 1999*). These technologies aim to monitor a system's state in real-time, respond promptly to emerging security issues, and ensure operational stability and safety.

Currently, design-time security assurance technologies are limited by the high cost when system reliability needs to be verified and required to deal with the complexity and uncertainty of a system (*Ferreira et al., 2010*), leading to a significant workload for verification. However, runtime security assurance technologies face challenges, such as balancing security and performance, need to manage complex system coordination and switching, and ensuring real-time system responses (*Desai et al., 2019*).

To address these issues, this article focuses on the design-time security assurance domain of UASs, aiming to improve development, verify efficiency, and reduce costs. The technical foundation of this research lies in architecture description languages (*Medvidovic & Taylor, 2000*) and formal verification techniques (*Clarke & Wing, 1996*). Architecture description languages describe and represent system architectures, providing an abstract, high-level means to express system components, properties, and behaviors, supporting system design and analysis. Architecture Analysis and Design Language (AADL) (*Feiler, Gluch & Hudak, 2006*) is a commonly implemented architecture description language. Formal verification techniques aim to verify the correctness of software systems, hardware circuits, or protocols through rigorous mathematical methods, helping developers pinpoint potential errors and security vulnerabilities. This article proposes an integrated process for modeling, verification, and code generation of UASs by combining architecture description languages and formal verification techniques. By synchronizing modeling and verification, the proposed method assists developers in pinpointing potential system vulnerabilities during the design phase of a system model, improving system reliability and decreasing verification costs. The code generation method then translates the verified models into flight controller code, thus enhancing development efficacy.

The main contributions of this article are summarized as follows:

- Based on the AADL, the article develops a set of generic UAS models that include the standard components and properties of UASs and can facilitate the design and verification of uncrewed systems.
- To use these models, the article specifies and verifies common safety properties of UASs and designs verification algorithms for two safety functions. Verification can pinpoint violations of specifications within a system and output specific counterexamples, aiding developers in identifying and remedying vulnerabilities.

- For the constructed models, the article introduces a method to generate flight controller code for UASs, which can directly produce viable flight controller code files from model files.

The structure of the remainder of this article is as follows: "Related research" introduces the related research. "Overall framework of the method" presents the overall framework of the proposed method. "UAS modeling", "The verification of safety properties and functions of UASs", and "Analysis and verification results" introduce the three main parts of the process, respectively. "Comparison Results" compares the proposed method with other available ones. A summary and future outlook are provided in "Summary and future work".

## RELATED RESEARCH

Uncrewed Aerial Systems (UASs), as complete physical information systems, are exposed to uncontrolled operational environments and face serious security threats. To ensure the safety and reliability of uncrewed systems, extensive research has been conducted by both academia and industry. *Dai et al. (2021)* developed an automatic testing platform for autopilot systems of UASs based on the field-programmable gate array (FPGA) hardware-in-the-loop simulation and model-driven design, thus capable of simulating various failure modes and scenarios to verify and assess the accuracy and credibility of simulation models. *Shafiee et al. (2021)* proposed a semi-quantitative reliability analysis framework based on the FMEA to pinpoint and assess the severity of failures in UASs during missions. *Sadhu, Zonouz & Pompili (2020)* introduced a deep learning-based approach for detecting and identifying the causes of UAS failures by analyzing inertial measurement unit sensor data to detect abnormal behaviors and identify the causes of failures. Several studies have explored the application of formal verification techniques in uncrewed systems. One of them was conducted by *Luckcuck (2023)*, who emphasized the role of formal methods as a key tool for ensuring the safety and correctness of uncrewed systems and believed that advancements in formal verification are crucial for addressing the challenges of verifying uncrewed systems. *Khan et al. (2020)* employed the formal proof tool Coq (*Shafiee et al., 2021*) to verify key hardware components of uncrewed systems, highlighting the significance of formal verification of hardware components, especially in implementations such as UASs and other computer-controlled systems.

The High-Assurance Cyber Military Systems (HACMS) project (*Cofer et al., 2017*) initiated by the Defense Advanced Research Projects Agency (DARPA) aims to research technologies for building highly reliable cyber-physical systems. The project developed a high-security UAS whose safety was proven by employing formal methods and effectively defended against malicious attacks from insiders. However, the project's report (*Fisher, Launchbury & Richards, 2017*) also mentioned its limitations, such as the significant workload and high costs in terms of time and required workforce for verification. The report also noted that verifying available code is more complex than simultaneous code development and verification, highlighting the need for research on formal verification methods during the system's development phase.

Among the presented studies above, methods based on in-the-loop simulation testing (*Dai et al., 2021*) and the FMEA (*Shafiee et al., 2021*) struggle to provide comprehensive reliability proofs; the reliability of machine learning-based fault detection methods (*Sadhu, Zonouz & Pompili, 2020*; *Taimoor et al., 2023*) in critical scenarios needs to be enhanced, and while formal method-based verification (*Luckcuck, 2023*; *Khan et al., 2020*; *Fisher, Launchbury & Richards, 2017*) can demonstrate system safety, the cost of verification is high. This article addresses these issues by proposing a method that balances reliability with efficiency.

## OVERALL FRAMEWORK OF THE METHOD

This section introduces the overall framework of the proposed method, as illustrated in Fig. 1.

The integrated method for modeling, verification, and code generation of UASs consists of three main components: UAS modeling, verification of safety properties and functions, and the code generation of the flight controller.

- In "UAS modeling", this article establishes a set of system models for UASs based on the AADL. The models, organized from top to bottom into multiple levels, can describe the overall structure and component properties of UASs. These models effectively support system design and verification. The generic models can also be applied to specific UAS models by modifying and refining certain components.

- This article builds upon the constructed system models to verify safety properties and functions. Within the OSATE (Open Source AADL Tool Environment) (*Feiler, Gluch & Hudak, 2006*), the AGREE (Analysis for GNU Real-Time Extension) language (*Whalen et al., 2012*) is employed to describe common safety property specifications for UASs. For frequently implemented functions in UASs, such as instruction encryption and GPS data security, the resolute verification framework (*Gacek et al., 2014*) is employed to design verification algorithms. After specifying the contracts, this article verifies the safety properties and functions, revealing potential vulnerabilities in the system.

- In "The code generation of the flight controller", this article investigates the transformation rules from the AADL code to the flight controller code and designs the conversion method. This method enables the transformation of the AADL system models into C++ code that can be utilized in the PX4 flight controller system (*Meier, Honegger & Pollefeys, 2015*).

The work mentioned above is implemented in the OSATE to support the AADL. The OSATE provides a complete toolkit for the AADL, including an editor, analyzer, and code generator, to support the design, analysis, and implementation of the AADL models (*Feiler, 2004*). Both AGREE and Resolute are formal verification tools integrated within the OSATE. These tools offer assertion languages for verifying system safety and functionality, with Resolute being more suitable for scenarios that require detailed proofs.

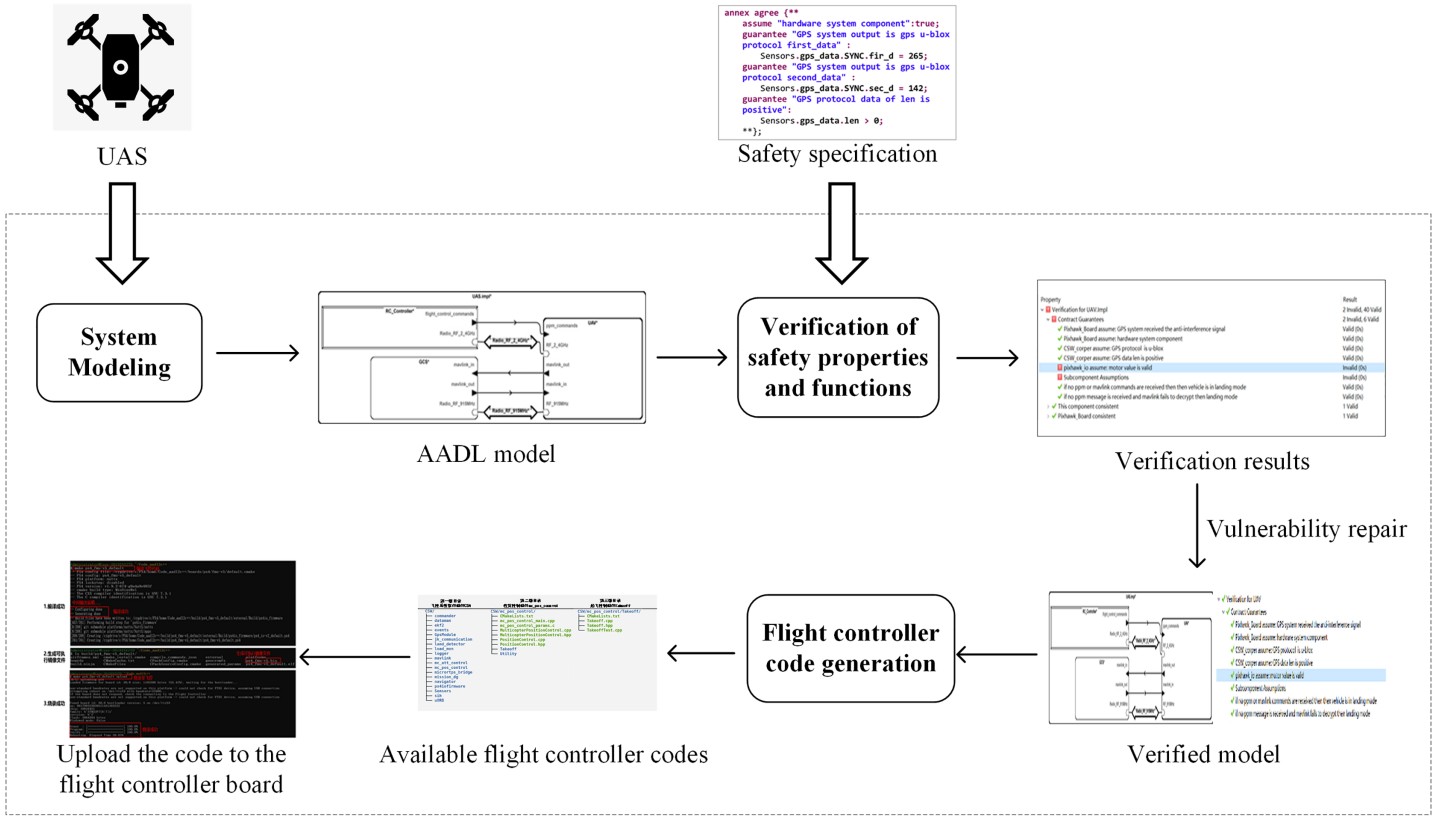

**Figure 1 Overall framework of the method.**

## Benefits of the proposed method

A clear benefit of this approach lies in its integration of modeling, verification, and automated code generation using AADL, which contrasts with traditional methods that often treat these phases as separate, manual processes. This integrated workflow not only enhances design efficiency by automating code generation from verified models but also significantly reduces the risk of introducing errors during manual transitions between design, verification, and implementation phases. By enabling early detection of potential vulnerabilities in the design phase and producing reliable, safety-compliant flight controller code directly from these verified models, the approach offers a streamlined, cost-effective, and error-resistant pathway for developing high-reliability UASs, distinguishing it from conventional methods.

## UAS MODELING

This section introduces the modeling method for UASs, beginning with analysing UAS architecture and then proceeding to model a system using the AADL.

### Architecture analysis of UASs

UASs are integrated systems that comprise a drone, ground station, and the corresponding communication links. The drone is an aircraft without an onboard pilot, capable of

autonomous flight or operation *via* remote control. The ground station is responsible for sending control commands to a drone and receiving data from it. UASs also require systems such as control and communication and the necessary equipment and personnel to control a flight (*Gupta, Ghonge & Jawandhiya, 2013*).

To facilitate modeling, this article categorizes the top-level components of the UASs into the ground station component, remote controller component, and flight controller system component. The ground station component communicates with the drone's flight controller system through a specific protocol, while the remote controller sends signals *via* a receiver to the flight controller system. These two components are primarily responsible for communication with the flight controller system. Therefore, we mainly describe their communication methods, connection relationships, and data interaction relationships without further refinement in the modeling. The focus of our modeling refinement is directed to the components of the flight controller system.

The flight controller system components include hardware, software, and output control components. The hardware component describes the physical composition that ensures a normal operation of the drone's flight control, including processors, sensors, power units, and GPS devices. The software component primarily describes the various process modules of the flight controller system, including software behaviors that respond to and process hardware devices, such as attitude control, position control, and fusion algorithms. The output control component mainly describes the control output process, such as converting pulse width modulation (PWM) control signals to electronic speed controllers and motors.

## The results of the AADL modeling

Based on the drone flight controller system described in the previous subsection, a UAS can be modeled by employing the AADL. Its design objective is to support multidimensional analysis of systems, including performance, safety, and reliability. The AADL can appropriately ignore the specific implementations of components, modeling and validating embedded systems by describing the attributes of elements and the interaction relationships between them. The AADL can also model systems and components with a hierarchical structure at any necessary level of detail to assess various aspects of system performance (*Feiler, Gluch & Hudak, 2006*).

This article models a system by describing the components at each layer of the drone system. In the model, the physical devices are abstracted into device components, and the data interfaces provided by each physical device are represented as ports in the AADL. The software is placed within a system component, which contains multiple processes. The top-level component structure of the system model constructed in this article is illustrated in Fig. 2.

The top-level component consists of uncrewed aerial vehicle (UAV), ground control station (GCS), and RC_Controller. UAV represents the drone's flight controller system model, GCS is the ground control station system model, and RC_Controller is the drone's remote controller model. These three components communicate through specific signal lines.
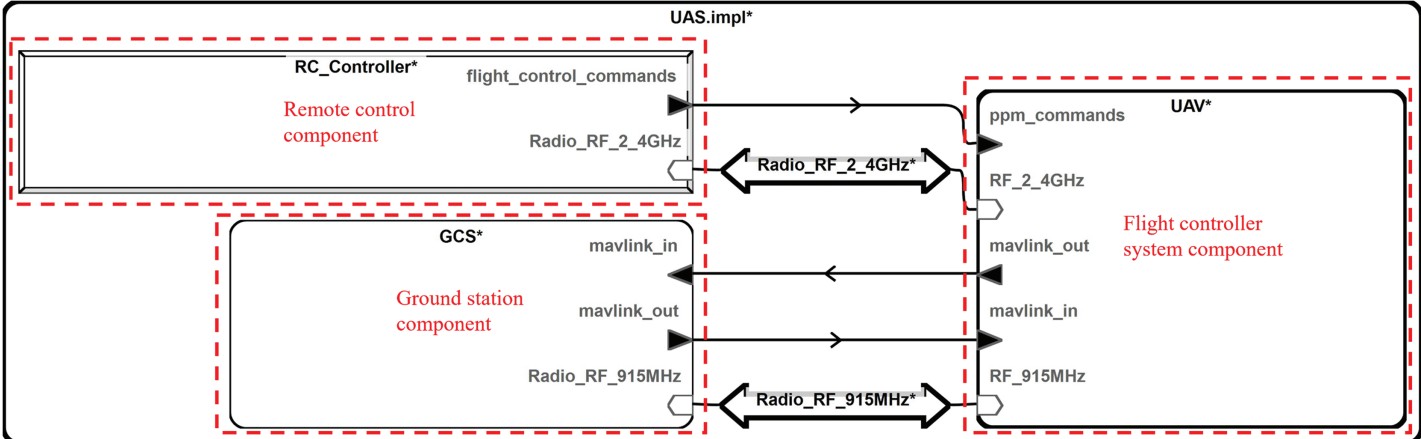

**Figure 2 Top-level component structure.**

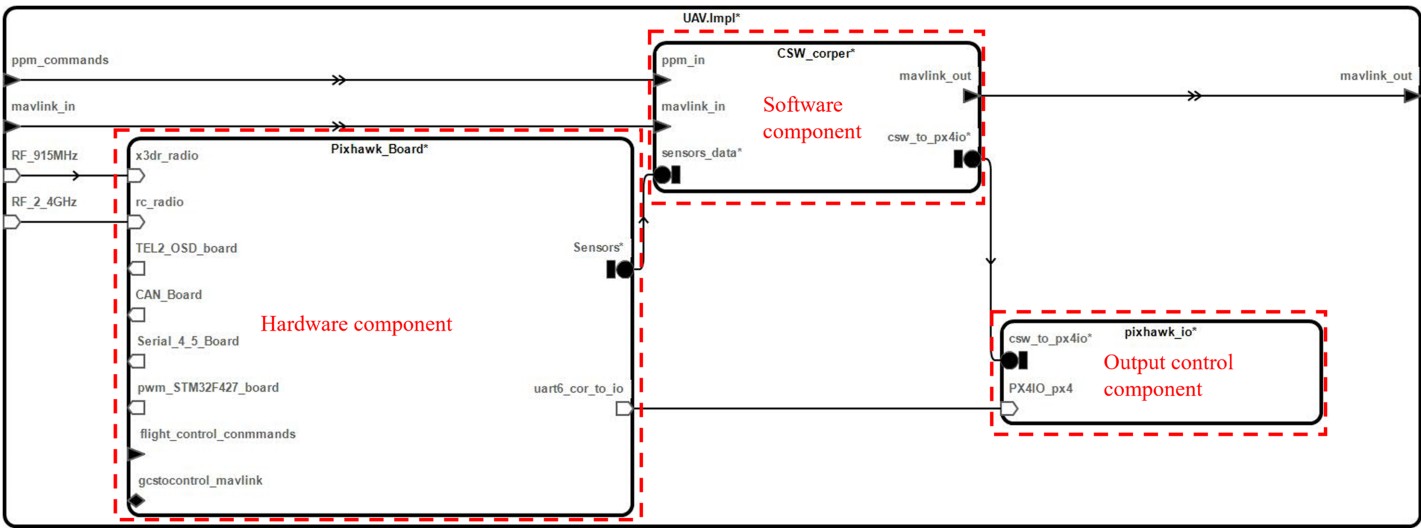

**Figure 3 Flight controller system component structure.**

The AADL supports multi-level modeling, and the composition of the flight controller system component of a UAV is illustrated in Fig. 3 at the next level.

The component of the flight controller system further comprises sub-components such as the flight controller system hardware component (Pixhawk_Board), software component (CSW_corper), and I/O board component (pixhawk_io). The representation of different inputs to the supply components is shown on the left side of Fig. 3. On the right-hand side, the outputs are depicted.

This article has completed the UAS's modeling through this layered modeling approach, with the model's overall structure illustrated in Fig. 4.

**Figure 4  Model overall structure diagram.**

```
processor ARM_Cortex_M7
features
bus_access: provides bus access;
properties
scheduling_protocol => (rms);
end ARM_Cortex_M7;
```

**Figure 5** **Subcomponent ARM_Cortex_M7 properties and interfaces.**

The overall structure diagram shows the components at various system levels and the sub-components they contain. The model has further refined components, but only up to the fourth level are listed here.

We have also defined each component's attributes, interfaces, *etc*. The sub-component ARM_Cortex_M7 in the processor component is taken as an example. As illustrated in Fig. 5, the above code describes a processor component named ARM_Cortex_M7, including its interface and attribute definitions.

In the AADL, the keyword processor represents components that execute computational tasks. The keyword features define the interfaces of the component, with bus access indicating that the processor provides bus access capabilities, enabling data transfer and communication. The keyword properties define the component's attributes, with scheduling_protocol being a scheduling policy attribute and rms being a commonly implemented scheduling strategy in real-time systems. It determines priorities based on the task's period, with shorter-period tasks having higher priority.

Using this model for design and verification offers the following advantages:

– The model includes commonly utilized components and attributes of UASs, ensuring generality. Developers can modify or refine some components to adapt to specific configurations of UASs, thereby shortening the development cycle.

– Based on this model, the components can be verified separately at various system levels. In the face of vulnerabilities, modular repairs can be made. The system's overall functionality can also be verified through the top-level components.

# THE VERIFICATION OF SAFETY PROPERTIES AND FUNCTIONS OF UASS

This section introduces the methods for verifying the safety properties and functions of UASs and separately describes specifications of safety properties and verification algorithms for security functions, followed by running experiments on verification and analysis.

```
annex agree {**
    assume "hardware system component":true;
    guarantee "GPS system output is gps u-blox
    protocol first_data" :
        Sensors.gps_data.SYNC.fir_d = 265;
    guarantee "GPS system output is gps u-blox
    protocol second_data" :
        Sensors.gps_data.SYNC.sec_d = 142;
    guarantee "GPS protocol data of len is
    positive":
        Sensors.gps_data.len > 0;
**};
```

**Figure 6** **Security property specification example.**

## A specification of safety properties

This article employs the AGREE language to specify the safety properties of UASs. The AGREE is an analysis tool based on the AADL model, supporting the formal description and verification of system properties within the AADL models. The AGREE is well-suited for analyzing critical attributes of real-time systems, such as task scheduling, response time, and resource allocation. Besides, it helps ensure a system operates correctly under specific constraints, improving system reliability (*Whalen et al., 2012*).

Figure 6 shows a safety property specification code for the hardware component of a flight controller system.

In the specification, three constraints are declared to which the system must adhere. The GPS data structure's first and second fields must be 265 and 142, respectively, following the GPS u-blox protocol. Additionally, the length field of the GPS data structure must be greater than 0. During system runtime verification, a search is conducted for the above data under various conditions. If counterexamples that do not meet the specifications are found, they are reported as vulnerabilities.

Similarly, the research describes common safety properties of UASs, as illustrated in Fig. 7. This article specifies safety properties at different levels of system components. Although it is challenging to consider all the necessary safety properties for a UAS during a design phase, this method allows for an incremental addition of required safety properties to each component, thereby improving system reliability.

## Verification algorithms for safety functions

This subsection verifies the instruction encryption and GPS data security functions of UASs and designs corresponding verification algorithms.

The implementation of the verification algorithms employs the Resolute framework, which is a tool for verifying the AADL models. Resolute provides an assertion language that allows developers to express system properties and constraints and supports logical proofs of these properties. Resolute's primary purpose is to rigorously assess the key properties of a system during the design phase, helping to ensure the reliability and safety

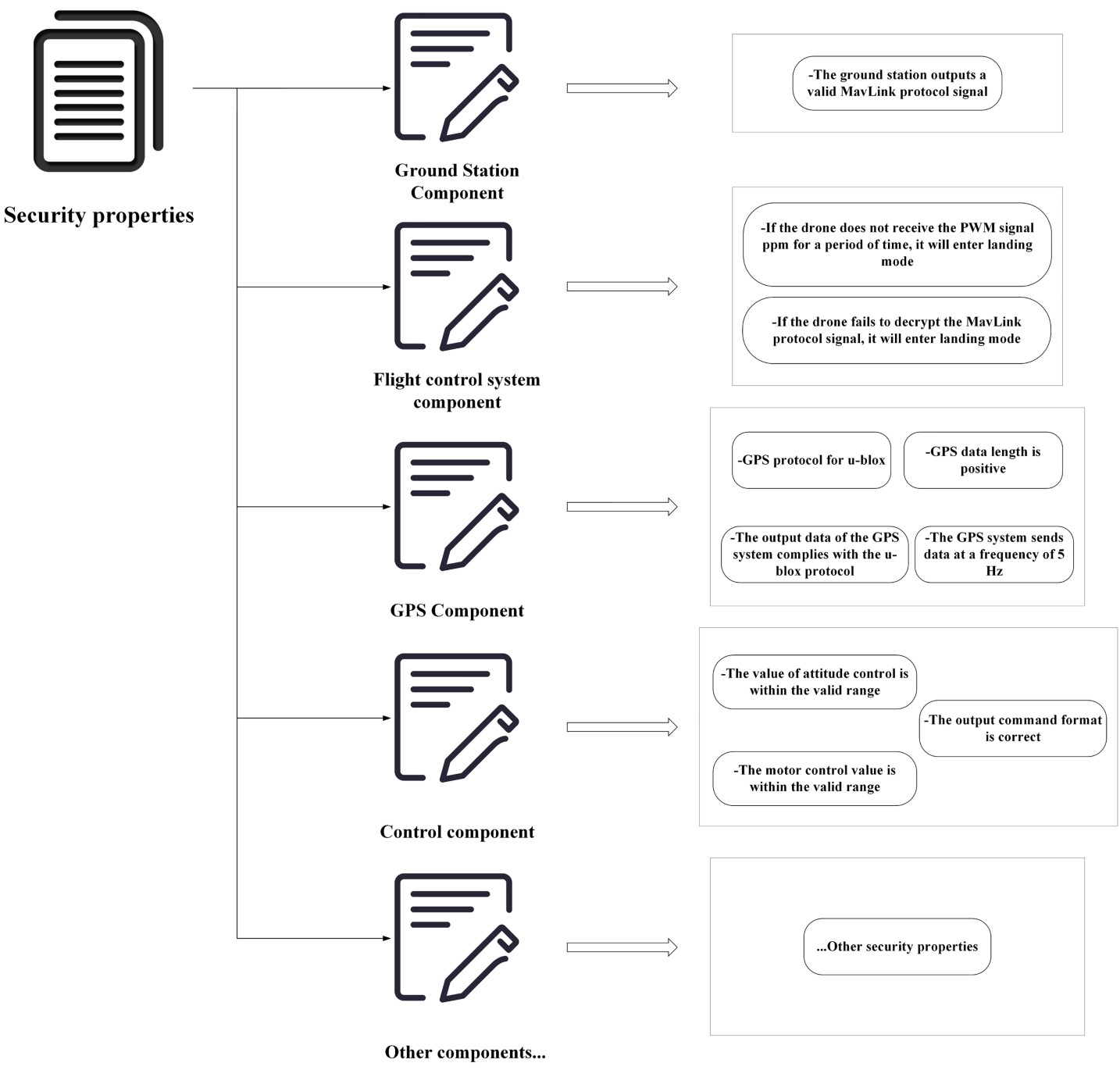

**Figure 7 Description of safety properties of UAS.**

of embedded systems (*Gacek et al., 2014*). Compared to AGREE, Resolute has stronger capabilities for logical proof, making it more suitable for scenarios requiring detailed proofs and logical validation.

Verification Algorithm for Instruction Encryption Function: The verification algorithm for the instruction encryption function mainly describes a mechanism for secure

instruction protection. It ensures that the motor components accept only encrypted instructions, preventing unauthorized access or tampering. Specifically, this mechanism includes the following steps:

– Instruction Verification: First, check that the instruction is from the ground station and has been encrypted.
– Verification of Encryption Algorithm: Validate the effectiveness of the encryption algorithm.
– Motors Only Accept Encrypted Instructions: Traverse each motor interface to ensure that the motors only accept and execute encrypted instructions.

The specific description of the algorithm is shown in Table 1.

Verification Algorithm for GPS Data Security Function: This algorithm aims to verify the security of GPS data, ensuring the integrity and security of the data during transmission and processing. The specific steps include:

– Input/Output Validity Check: Verify if the trajectory history and output history are linear to ensure the continuity and consistency of the data.
– GPS Data Verification: Check each input data of the software component.

Ensure that all GPS data reaching the software component is encrypted. The specific description of the algorithm is shown in Table 2.

## ANALYSIS AND VERIFICATION RESULTS

After completing the system modeling, safety property specifications, and functional verification algorithms, we executed the verification using the Z3 solver integrated within the OSATE. Several specification violations were discovered throughout the verification process, as shown in Table 3. After discovering vulnerabilities, the reasons for these vulnerabilities can be analyzed within the OSATE. To take the specification "The value of motor control is within the effective range" as an example, the verification output results are illustrated in Fig. 8. The OSATE also provides a feature for tracing counterexamples, as shown in Fig. 9. For the specification violated in Fig. 7, it can be traced that the verification fails when the value of csw_to_px4io.motor0- 5. factor (the pulse width modulation signal duty cycle for motor 0) is 1503.

In the specification, the range of this variable is limited to 1,000–1,500, which is violated in this case. To resolve this vulnerability, we traced back to the location in the control component where the variable is assigned its value. We found that the range of the assignment for this variable is 1,000–2,000 within the control component, which is inconsistent with the specification. By correcting this range, the vulnerability can be fixed. Similar approaches can be employed to resolve other vulnerabilities in the system.

Experiments have shown that modeling and verifying a UAS can help developers efficiently discover hidden vulnerabilities during a design phase, ensuring that the system meets the specifications of the specified safety property.

**Table 1 Verification Algorithm for Instruction Encryption Function.**

**Procedure:** Instruction_safety *%Instruction safety function verification*

**Input:** CSW::CSW, Hardware_IO::motor *%Input software component and motor component*

**Output:** verification result *%Output verification result*

1. **Main**(CSW::CSW, Hardware_IO::motor) *%Main function*

    main_loop <- instance(CSW::CSW); *%Define the main loop component main_loop as an instance of the software component*

    motors <- instances(Hardware_IO::motor); *%Define the motors component motors as an instance of the motor component*

    **If** only_gs_encrypt() **and** motors_only_receive_decrypt(main_loop, motors) **then:**

    *% If only the ground station can send encrypted instructions and the motors only accept encrypted instructions, then verification passes*

        **Return** True;

    **Else:**

        **Return** False;

2. **only_gs_encrypt**() *%Verify that only the ground station can send encrypted instructions*

    algo1 <- property(CSW::CSW, px4_sys::Encryption_Algorthim);

    algo2 <- property(px4iofirmware::px4iofirmware, px4_sys::Encryption_Algorthim);

    *%Obtain the ground station encryption algorithm algo1 and the flight control firmware encryption algorithm algo2*

    **If** algo1 == algo2 **and** authenticated_encryption(algo1) **and** private_key() **then:**

    *%If the two encryption algorithms are the same, the encryption algorithm is strong, and a private key is used, then verification passes*

        **Return** True;

    **Else:**

        **Return** False;

3. **authenticated_encryption**(e) *%Verify that the encryption algorithm is a strong encryption algorithm*

    **If** e == "AES-128-GCM" **or** e == "AES-128-CTR-HMAC-SHA1" **then:**

    *%If the encryption algorithm is one of the two specified strong encryption algorithms, then verification passes*

        **Return** True;

    **Else:**

        **Return** False;

4. **motors_only_receive_decrypt**(main_loop, motors) *% Verify that the motors only accept encrypted instructions*

    **For** each motor m **in** motors **do**: *% Iterate over each motor*

      **For** each c **in** connects **do**: *% Iterate over all connections of the motor*

        **If** is_data_port_connection(c) **or** is_event_data_port_connection(c) **then:**

        *% Verify that the motor data port or event data port only receives instructions from the main loop component*

          **If** comes_from_mainloop(c, main_loop) **then:**

            **Continue;**

          **Else:**

            **Return** False;

    **For** each data **in** main_loop **do**: *% Iterate over the data of the main loop component*

      **If** is_incomming(data, main_loop) **then:** *% If the data is input data of the main loop component*

(*Continued*)

**If** is_sensor_data(data) **or** feature_confined_to_decrypt(data, main_loop) **then:**

*% If all data is either sensor data or encrypted data, pass verification*

**Continue;**

**Else:**

**Return** False;

**Return** True;

---

**Table 2 Verification algorithm for GPS data security function.**

**Procedure :** Gps_data_safety *%GPS data safety function verification*

**Input:** CSW::CSW *%Input software component*

**Output:** verification result *%Output verification result*

1. **Main**(CSW::CSW) *% Main function*

 main_loop <- instance(CSW::CSW);

 *% Define the main loop component main_loop as an instance of the software component*

 **If** input_and_output_of_model_are_valid() **and** input_checked_GPS() **then:**

 *% If the input and output data are valid and all input GPS data is encrypted, then verification passes*

  **Return** True;

 **Else:**

  **Return** False;

2. **input_and_output_of_model_are_valid**() *% Verify that the input and output data are valid*

 **If** history_trajectory_is_liner() **and** history_outputs_are_liner() **then:**

 *% If the historical trajectory of the data is linear and the outputs are linear, then verification passes*

  **Return** True;

 **Else:**

  **Return** False;

3. **input_checked_GPS(main_loop)** *% Verify that all input GPS data is encrypted*

 **For** each data **in** main_loop **do:** *% Iterate over the data of the main loop component*

  **If** is_incomming(data, main_loop) **then:** *% If the data is input data of the main loop component*

   **If** is_GPS_data(data) **And** feature_confined_to_decrypt(data, main_loop) **then:**

   *% If all GPS data is encrypted, then verification passes*

    **Continue;**

   **Else:**

    **Return** False;

 **Return** True;

**Table 3 Vulnerability list.**

| Vulnerability | Component | Specification |
|---|---|---|
| GPS data security | GPS component | property(a, gps_property::Is_Checked) =true |
| The motor control value is within the valid range | Control component | csw_to_px4io.motor0.dutyfactor >= 1000 and csw_to_px4io.motor0.dutyfactor <= 1500 |
| Sensor data accuracy meets standards | Sensor component | sensor_data_precision > 0.1 |
| The bus bandwidth is higher than the data transfer rate | Bus component | bus_bandwidth > data_rate |
| The value of attitude control is within the valid range | Attitude control component | ValidOutputRange.contains(Att_Control_Output) |

| Property | Result |
|---|---|
| ⌄ 🔲 Verification for UAV.Impl | 3 Invalid, 39 Valid |
|   ⌄ 🔲 Contract Guarantees | 2 Invalid, 6 Valid |
|     ✔ Pixhawk_Board assume: GPS system received the anti-interference signal | Valid (0s) |
|     ✔ Pixhawk_Board assume: hardware system component | Valid (0s) |
|     ✔ CSW_corper assume: GPS protocol is u-blox | Valid (1s) |
|     ✔ CSW_corper assume: GPS data len is positive | Valid (1s) |
|     🔲 pixhawk_io assume: motor value is valid | Invalid (0s) |
|     🔲 Subcomponent Assumptions | Invalid (0s) |

**Figure 8 Validation results.**

| pixhawk_io | | |
|---|---|---|
| pixhawk_io..ASSUME.HIST | TRUE | FALSE |
| pixhawk_io.csw_to_px4io.motor0.dutyfactor | 1000 | 1503 |

**Figure 9 Counterexample tracking.**

### The code generation of a UAS flight controller

This subsection introduces the methods and experiments for generating flight controller codes for UASs.

#### Code generation method

The code generation method involves a layer-by-layer transformation of the components. The software components modelled include the system components, process, thread, data, and sub-routine. This subsection primarily discusses the rules for converting each type of component.

System Component: In the AADL, the declaration of the system components is illustrated in Fig. 10, and the system components should include both declaration and implementation.

```
 -- Declarations of system components
system systemTypeID
end systemTypeID;
-- Implementation of system components
system implementation systemTypeID.impl
end systemTypeID.impl;
```

**Figure 10 Declaration of system components.**

The conversion rules for system components are as follows: System components are transformed into a folder named systemTypeID. This folder must contain at least one systemTypeID.hpp file, which includes declarations of global variables and functions and contains common C and C++ language header files. An identifier name, systemTypeID, is used for the system type declaration. All code files generated from the subcomponents of the system component are placed within the systemTypeID folder. This folder contains not only the code files generated from the system component but all the code files generated from its subcomponents. For subcomponents, the system component will generate corresponding code files based on the specific conversion rules of the subcomponents.

Process Component: The conversion rules for process components are as follows: Each process subcomponent within the system component is transformed into a pair of processTypeID.hpp and processTypeID.cpp files. "processTypeID.hpp" contains declarations of data and functions shared by threads within the process and includes a reference to the systemTypeID.hpp file. "processTypeID.cpp" contains the implementations of the data and function declarations and includes the entry functions for the subcomponents of the process component: thread components and subroutine components. The conversion mapping is shown in Fig. 11.

In the context of the code generation for UAS flight control systems, the function Fun_processTypeID() is the entry point for process tasks. CreatePeriodicTask() is employed to create a periodic task, with TaskName being the task's name, TaskEntryPoint_threadTypeID() being the entry function of the task, and status representing the task's status. Fun_threadTypeID() is the entry function for thread tasks within the process, indicating that the thread subcomponent task is executed once the process task is initialized.

Thread Component: The conversion rules for thread components are as follows: Each thread subcomponent of the process component corresponds to a pair of threadTypeID.hpp and threadTypeID.cpp files. "threadTypeID.hpp" contains declarations of data and subroutines within the thread and includes the processTypeID.hpp file. "threadTypeID.cpp" contains the data implementations and subroutine declarations. The conversion mapping is shown in Fig. 12.

Subroutine Component: The conversion rules for subroutine components are as follows: Subroutine components are transformed into callable functions, and the generated

| AADL framework | Generated code framework |
|---|---|
| process **processTypeID** <br> -- Declaration of process components <br> **end processTypeID;** <br> **process implementation processTypeID.impl** <br> -- Implementation of process components <br> **end processTypeID.impl;** | **#include "processTypeID.hpp"** <br> /* Contains declarations of data and functions, and references to the systemTypeID.hpp */ <br> **int Fun_processTypeID()** <br> { <br>   CreatePeriodicTask(TaskName, <br>   TaskEntryPoint_threadTypeID, status); <br>   /* Creating periodic thread task */ <br>   **return 1;** <br> } <br> **int TaskEntryPoint_threadTypeID()** <br> { <br>   Fun_threadTypeID(); <br>   /* Implementation of thread tasks */ <br>   **return 1;** <br> } |

**Figure 11 Process component conversion rules.**

| AADL framework | Generated code framework |
|---|---|
| **thread threadTypeID** <br> -- Declaration of thread components <br> **end threadTypeID;** <br> **thread implementation threadTypeID.impl** <br> -- Implementation of thread components <br> **Properties** <br>   Dispatch_Protocol => Periodic; <br>   -- Set the thread attribute to periodic thread <br>   Period=>100ms; <br>   --Set the period value <br> **end threadTypeID.impl;** | **#include "threadTypeID.hpp"** <br> /* Contains declarations of thread data and subroutines, and references to the processTypeID.hpp */ <br> **int Fun_threadTypeID ()** <br> { <br>   /* Implementation of data and subroutines within threads */ <br>   **return 1;** <br> } |

**Figure 12 Thread component conversion rules.**

code is included within the process component or thread component in which the subroutine resides. The conversion mapping is shown in Fig. 13.

In this context, Fun_tSporadicSync() is the entry point of the task, and tSporadicSync_spClient() is the corresponding function for the subroutine, which can also include parameters.

| AADL framework | Generated code framework |
|---|---|
| **thread implementation threadTypeID.impl** <br> --The call to the subroutine is contained in the implementation of the thread component <br> **calls{** <br> spClient: subprogramTypeID.impl; <br> } <br> --Declare the mode of subroutine call <br> **end threadTypeID.impl;** <br> -- Declaration of subroutine components <br> **subprogram subprogramTypeID** <br> **end subprogramTypeID;** <br> --Implementation of subroutine components <br> **subprogram implementation subprogramTypeID.impl** <br> **end subprogramTypeID.impl;** | /* The code generated by the subroutine component is contained in the process component or thread component */ <br> **int Fun_tSporadicSync**() <br> /* Task entry point */ <br> { <br> **tSporadicSync_spClient();** <br> /* Calling the subroutine implementation function */ <br> **return 1;** <br> } <br> **int tSporadicSync_spClient**() <br> { <br> /* Subroutine Implementation */ <br> **return 1;** <br> } |

**Figure 13 Subroutine component conversion rules.** 

| AADL framework | Generated code framework |
|---|---|
| **data bool** <br> -- Definition of Boolean type data component <br>   **properties** <br>   Data_Model::Data_Representation <br>   => Boolean; <br> **end bool;** <br> **data int** <br> --Definition of integer type data components <br>   **properties** <br>   Data_Model::Data_Representation <br>   => Integer; <br> **end int;** | **bool dataInstanceName;** <br> /* Boolean data */ <br> **int dataInstanceName;** <br> /* Integer data */ |

**Figure 14 Data component conversion rules.** 

    Data Component: The conversion rules for data components are as follows: Data components are transformed into the corresponding data types in C++, and data subcomponent instances are converted into variables of the corresponding type. The code generation occurs within the process or thread component files that contain the data components. The conversion mapping is shown in Fig. 14.

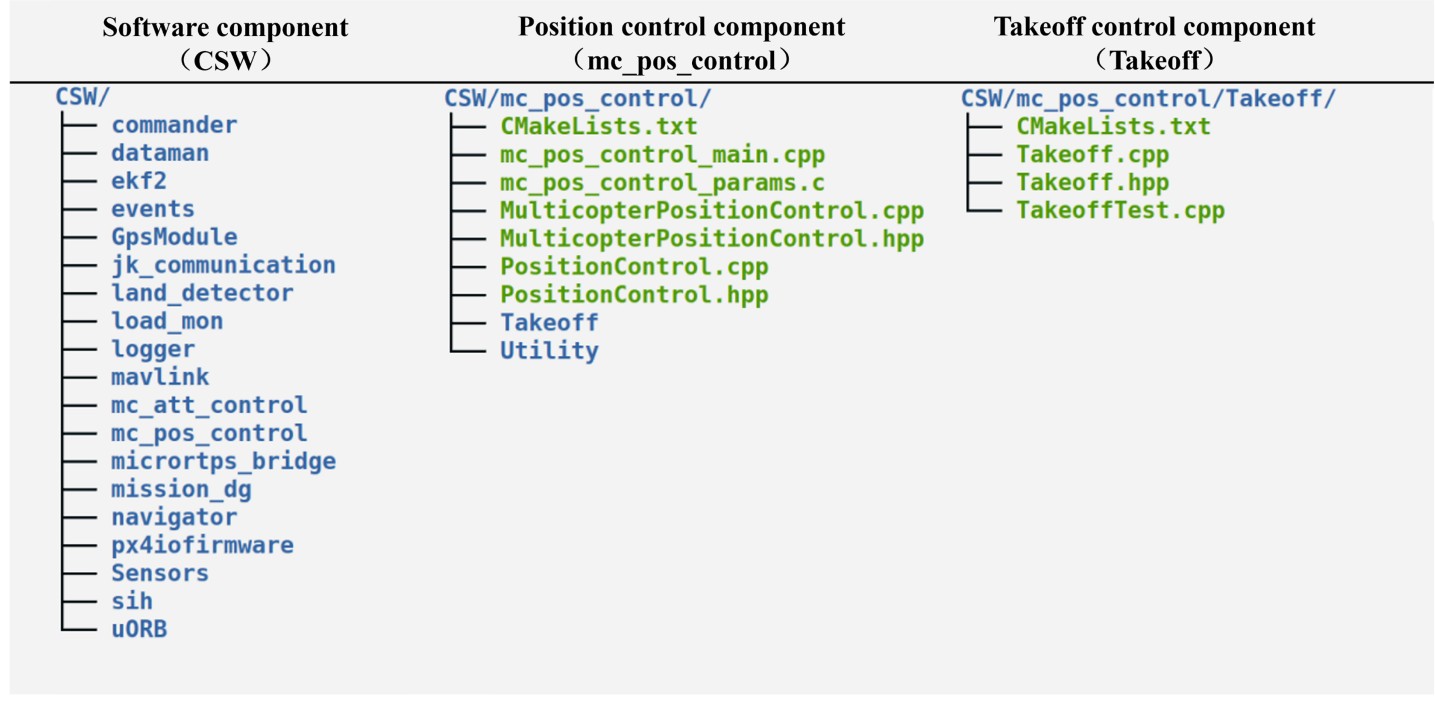

**Figure 15 Directory structure of generated flight controller code.**

Conversions involve features, connections, flows, and other AADL elements during the component transformation process. Corresponding conversion rules have been designed for these elements, with the approach being similar to the above and thus will not be elaborated on here.

## The experiment of the code generation

This subsection performs an experiment using the code presented above based on the generation method to transform the UAS model. After the transformation is run, the directory structure of the generated flight controller code is shown in Fig. 15.

The directory structure matches the structure of the software component in the UAS model.

To verify the correctness of the generated code, we successively tested whether the code could be compiled successfully, generated executable image files, and uploaded to the PX4 flight controller board. The output of the test is shown in Fig. 16.

As illustrated, the code was compiled correctly, generating an executable image file px4_fmu-v5.bin and successfully uploading it to the flight controller board. This code is also employed to perform drone simulation flights, and during the simulation, the system could function normally, indicating that the generated code is viable.

### *Scalability and real-world usage scenarios*

The scalability of the proposed approach—integrating modeling, verification, and code generation for UASs—can be examined across several dimensions: adaptability to different

**Figure 16 Generated code test.**

UAS types, flexibility with increasing complexity, and extensibility to various operational environments. Here's how this approach supports scalability.

A few of the real-world examples where the proposed could be used are as follows: disaster response and emergency relief, industrial inspection in hazardous environments, autonomous delivery systems in urban environments, military surveillance and reconnaissance missions, and agricultural monitoring and crop management.

## COMPARISON RESULTS

We have compared the proposed method with the available design-time security assurance technologies for UASs. Table 4 summarizes the results.

Compared to other methods, the proposed method is more comprehensive and capable of system modeling, vulnerability detection, formal proof, and the code generation of the flight controller. Unlike in-the-loop simulation testing (*Dai et al., 2021*), Failure Modes and Effects Analysis (*Huang et al., 2020*), and machine learning-based vulnerability detection (*Sadhu, Zonouz & Pompili, 2020*), the proposed method employs formal techniques for verification, making the reliability proof more complete (*Clarke & Wing, 1996*). Compared to formal methods (*Luckcuck, 2023*), the proposed approach can detect vulnerabilities at a lower cost during the system modeling phase (*Fisher, Launchbury & Richards, 2017*). Additionally, the suggested process includes the capability to generate

**Table 4 Method function comparison.**

| Method | System modeling | Vulnerability detection | Formal proof | Flight controller code generation |
|---|---|---|---|---|
| UAV-in-the-loop simulation test (*Dai et al., 2021*) | ✓ | ✓ | | |
| Failure mode and effects analysis (*Huang et al., 2020*) | | ✓ | | |
| Vulnerability detection method based on machine learning (*Sadhu, Zonouz & Pompili, 2020*; *Taimoor et al., 2023*) | | ✓ | | |
| Formal methods (*Luckcuck, 2023*; *Khan et al., 2020*) | | ✓ | ✓ | |
| Our method | ✓ | ✓ | ✓ | ✓ |

flight controller code and transform verified system models into flight controller code to enhance development efficiency.

# SUMMARY AND FUTURE WORK

To address the challenges of high time and labour costs associated with designing and verifying highly reliable UASs, this article examines an integrated method for modeling, verification, and code generation of UASs. The suggested algorithm combines architecture description languages and formal techniques, offering the advantages of lower verification costs and higher development efficiency.

However, limitations exist to the current method, as UASs require numerous safety properties, which are difficult to describe through manual design during the design phase fully.

The next research phase involves leveraging large language model technology to generate safety property specifications automatically. The available studies discuss the use of large models for generating formal specifications (*Kogler, Falkner & Sperl, 2024*), such as using large models to generate safety specifications in the domain of smart contracts (*Liu et al., 2024*). Also, we plan to explore the use of large models to assist in generating safety specifications for UASs, thereby enhancing their reliability.

The proposed approach enhances safety assurance in UASs development through its integration of modeling, verification, and automated code generation. It can contribute to safety in the following ways.

**Early Detection of Design Flaws:** By using AADL for modeling, the approach enables early identification of potential vulnerabilities in the system. This pre-implementation detection minimizes risks associated with design flaws that might otherwise go unnoticed until later stages, making it easier and safer to address issues.

**Automated Code Generation from Verified Models:** The approach eliminates errors that could arise during manual coding by generating flight controller code directly from verified models. This automated process produces safer, more reliable code that reflects the rigorously tested model specifications, ensuring safety compliance throughout development.

**Consistency Across Development Phases:** Integrating modelling, verification, and code generation provides consistency across all development phases, reducing the

potential for discrepancies that can arise when different teams or tools are used separately. This continuity improves the reliability of the UAS software and ensures that safety protocols are maintained throughout the lifecycle.

**Dynamic Adaptability to Safety-Critical Scenarios:** With its flexible, model-driven design, the approach can be adapted for various safety-critical applications (*e.g.*, disaster response industrial inspection). The approach ensures that the UASs are well-prepared for real-world scenarios, maintaining high safety standards even under challenging conditions by testing and verifying safety features specific to each application.

**Reduction in Human Error:** Automating complex verification and code generation processes reduces human error, especially in safety-sensitive systems. The automated approach ensures rigorous standards are consistently applied, minimizing risks associated with manual oversights or inconsistencies.

### Funding
The authors received no funding for this work.

### Competing Interests
The authors declare that they have no competing interests.

### Author Contributions
- Jianyu Zhang conceived and designed the experiments, performed the experiments, prepared figures and/or tables, and approved the final draft.
- Long Zhang conceived and designed the experiments, performed the experiments, analyzed the data, performed the computation work, prepared figures and/or tables, authored or reviewed drafts of the article, and approved the final draft.
- Yixuan Wu conceived and designed the experiments, prepared figures and/or tables, authored or reviewed drafts of the article, and approved the final draft.
- Linru Ma conceived and designed the experiments, performed the experiments, analyzed the data, performed the computation work, authored or reviewed drafts of the article, and approved the final draft.
- Feng Yang performed the experiments, analyzed the data, performed the computation work, authored or reviewed drafts of the article, and approved the final draft.

### Data Availability
The project code is available in the Supplemental File.

The data is available at Zenodo: Khatiri, S., Saurabh, P., Zimmermann, T., Munasinghe, C., Birchler, C., & Panichella, S. (2024). SBFT 2024 UAV Test Case Generation Competition: Evaluation Artifacts [Data set]. Zenodo. https://doi.org/10.5281/zenodo.10605584.

## Supplemental Information

Supplemental information for this article can be found online at http://dx.doi.org/10.7717/peerj-cs.2575#supplemental-information.

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
