# Peer review of "An integrated modeling, verification, and code generation for uncrewed aerial systems: less cost and more efficiency"

_PeerJ Computer Science, doi:10.7717/peerj-cs.2575_

## Round 0.1 · original submission · Major Revisions

Dear authors

We have received experts opinion on your manuscript, you will see that the experts are suggesting couple of revisions before we reconsider your manuscript. I do agree with these improvement suggestions and also suggest you to consider the following comments along with others.

Please carefully revise your manuscript along with detailed responses

Additional Editor Comments: please improve the abstract to clarify the overall objective of the study.
Please mention /clarify the statement "based on the verified models." , which are these models?
Please improve the language quality of paper

·

Basic reporting

1. Add 1 or 2 line on Potential vulnerability.
2. Add 2 or 3 lines on AADL system in introduction.
3. Diagram 1 not properly visible.
4.Add table for results comparison

Experimental design

Add a diagram on experimental design.

Validity of the findings

No comments

Additional comments

This paper has been accepted with minor correction. Overall, the paper is strong and seems eligible for publishing.

Reviewer 2 ·

Basic reporting

The authors in this paper, “Integrated modeling, verification, and code generation for uncrewed aerial systems,” aim to introduce an integrated approach to overcome the challenge of designing and implementing highly reliable UAS while effectively controlling development costs and enhancing efficiency. The authors are suggested to address the following comments while revising the paper.

1: Title: Extend the title to include why this integration is being done.

2: Abstract: Include some key results/metrics on the evaluation and effectiveness of the proposed approach.

3: The paper starts with reference number [11,25], the authors are suggested to reorganize the reference in ascending order.

4: Reorganize the paragraphs in the introduction to follow a uniform structure and length.

5: Improve the resolution and readability of text for all Figures. Currently, the images are blur and the text is small.

6: Extend the related work section by adding more recent studies.

Experimental design

7: Clearly state the evaluation criteria to evaluate the effectiveness of the proposed approach and compare it with the approaches in the existing literature.

8: A discussion on the experimental results and evaluations seems missing. A detailed discussion should be provided on the results and findings.

9: Method Comparison should be discussed in more detail.

Validity of the findings

10: Rewrite the summary and future work section with more details on what is done, why it is done, how it is done what is achieved, and how it’s compared with the existing methods to report on the effectiveness.
Afterwards, in the second paragraph discuss the implications, limitations, and future work of the approach proposed in this paper.

Reviewer 3 ·

Basic reporting

The research investigates an integrated approach to improve the design and development of Unmanned Aerial Systems (UAS) by using Architecture Analysis and Design Language (AADL) for modeling, verification, and code generation. It proposes generic UAS models, defines formal specifications for safety and functionality. Overall the paper seems to have good contributions but I suggest the following improvements:

Please provide a clear benefit of how this approach differs from or improves upon existing methods. The innovation and specific advantages of this integrated approach need to be more explicitly stated to make the contribution more clear.
The paper lacks in details about detailed experiments. Please provide more information on the experimental setup, data, and analysis.
The paper would provide benefit from discussing the scalability of the proposed approach. UAS applications range from small, simple systems to highly complex ones. How well does this approach handle varying levels of system complexity, and can it be applied to larger, more intricate UAS?
Please try to incorporate real-world case studies or validation examples.

One of the key goals is to control development costs and improve efficiency, but the paper does not thoroughly explore how these are achieved or measured.
In a separate section , please discuss the potential limitations or challenges of implementing the proposed approach, such as the complexity of integrating this system into existing UAS etc.

Since UAS are used in safety-critical fields, the paper should place more emphasis on how the proposed approach improves safety assurance.

Experimental design

The paper lacks in experimental details.

Validity of the findings

The complete review has been addressed under the basic reporting.

---

## Round 0.2 · Minor Revisions

Dear authors,
The reviewers have now commented on the revised version of your manuscript. They are happy with most of the improvements, but still, a few important points need your attention. Therefore, we invite you to carefully update and resubmit

·

Basic reporting

An Integrated Modeling, Verification, and Code Generation for Uncrewed Aerial Systems: Less Costand More Efficiency
1. Clear and unambiguous
2. Sufficient reference
3. Professional structure of the paper

Experimental design

Every thing well defined and follow the technical ethical standards.

Validity of the findings

Literature is clearly stated and all underlined data provided.

Additional comments

Overall, this article is well-written and aligns with my views; therefore, I find it acceptable

Reviewer 2 ·

Basic reporting

no comment

Experimental design

no comment

Validity of the findings

no comment

Reviewer 3 ·

Basic reporting

The research investigates an integrated approach to improving the design and development of Unmanned Aerial Systems (UAS) by using Architecture Analysis and Design Language (AADL) for modeling, verification, and code generation. It proposes generic UAS models, defines formal specifications for safety and functionality, overall the paper seems to have good contributions but I suggest the following improvements.

Experimental design

Please provide a clear benefit of how this approach differs from or improves upon existing methods. The innovation and specific advantages of this integrated approach need to be more explicitly stated to make the contribution more clear.

The paper lacks details about detailed about experiments. Please provide more information on the experimental setup, data, and analysis.
paper would benefit from discussing the scalability of the proposed approach. UAS applications range from small, simple systems to highly complex ones. How well does this approach handle varying levels of system complexity, and can it be applied to larger, more intricate UAS?
Please try to incorporate real-world case studies or validation examples.

Validity of the findings

One of the key goals is to control development costs and improve efficiency, but the paper does not thoroughly explore how these are achieved or measured.
In a separate section, please discuss the potential limitations or challenges of implementing the proposed approach, such as the complexity of integrating this system into existing UAS etc.

Since UAS are used in safety-critical fields, the paper should place more emphasis on how the proposed approach improves safety assurance.

---

## Round 0.3 · accepted · Accept

Thank you for your revision, I'm pleased to inform you that your manuscript is being recommended for publication based on feedback from experts and my evaluation.
Thank you for your contribution